# Show Me More! The Influence of Visibility on Sustainable Food Choices

**DOI:** 10.3390/foods8060186

**Published:** 2019-05-31

**Authors:** Nicky Coucke, Iris Vermeir, Hendrik Slabbinck, Anneleen Van Kerckhove

**Affiliations:** BE4LIFE, Department of Marketing, Innovation and Organization, Faculty of Economics and Business Administration, Ghent University, 9000 Ghent, Belgium; Iris.Vermeir@UGent.be (I.V.); Hendrik.Slabbinck@UGent.be (H.S.); Anneleen.Vankerckhove@UGent.be (A.V.K.)

**Keywords:** choice architecture, sensory nudges, visual cues, sustainable consumer behavior, display area size, quantity of displayed products, visibility

## Abstract

Visual cues are omnipresent in an in-store environment and can enhance the visibility of a product. By using these visual cues, policy makers can design a choice environment to nudge consumers towards more sustainable consumer behavior. In this study, we use a combined nudge of display area size and quantity of displayed products to nudge consumers towards more sustainable meat choices. We performed a field experiment of four weeks in a butchery, located in a supermarket. The size of the display area and quantity of displayed poultry products, serving as the nudging intervention, were increased, whereas these were decreased for less sustainable meat products. In order to evaluate the effectiveness of our nudging intervention, we also collected data from a control store and performed a pre-and post-intervention measurement. We kept records of the sales data of the sold meat (amount of weight & revenue). When conducting a three-way ANOVA and post hoc contrast tests, we found that the sales of poultry increased during the nudging intervention, but did not decrease for less sustainable meat products. When removing the nudge again, the sales of poultry decreased again significantly in the experimental store. Changing the size of display area and the amount of products displayed in this display area created a shift in the consumers’ purchase behavior of meat.

## 1. Introduction

Throughout human history, consumer diets have always been characterized by a significant intake of meat [1,2]. The production of meat has experienced a huge rise in recent decades and will continue to grow rapidly in the future [2,3,4]. Food production, however, especially the production of meat, is the human activity with the single largest impact on the environment [5]. Meat production requires a lot of land, water, and feed to breed livestock, which makes this a very energy-intensive type of food [6,7]. Besides the use of large amounts of resources, meat production also releases a large amount of greenhouse gases (GHGs) into the environment [8,9]. These emissions are the cause of several environmental concerns such as climate change, the loss of biodiversity, and changes in the nitrogen cycle [8,10].

In response to this, policy makers are questioning the current global meat production and consumption patterns and are searching for ways to lead consumers towards more sustainable meat consumption. By sustainable meat consumption, we mean the consumption of meat products that have a lower impact on the environment based on the use of fossil energy during the production process, the use of land for breeding livestock, the generated water footprint (WF) and the emissions of GHGs produced during the entire production process [11,12,13,14]. Based on these elements, not all types of meat have an equally strong impact on the environment. Looking at the CO_2_ emissions (Type of GHG) from the production of white meat, such as poultry, it is noted that white meat has a much lower impact on the environment compared to types of red meat, including beef, lamb, and pork [8,15]. Lamb/sheep products have the highest impact on the environment based on CO_2_ emissions, with on average 20 CO_2_/kg, which is even more than the impact of beef, 15 CO_2_/kg. Compared to these types of meat, both pork (5 CO_2_/kg) and especially poultry (2 CO_2_/kg) have a much lower impact on the environment based on CO_2_ emissions [15]. The production of poultry also requires less land for breeding poultry, generates a lower water footprint and requires less fossil energy compared to the production of other types of meat, which again results in a lower environmental impact [8,16,17].

In order to encourage sustainable meat consumption, two paths can be taken [12,18]. First, the total amount of meat consumed can be lowered. This could be achieved be offering smaller portions of meat to consumers [19,20] or by offering more vegetarian alternatives. For consumers, however, eating less or even no meat at all could be a huge step to take and can be a very difficult cultural habit to break [4,6]. Based on this, we follow a second path in order to strive for a more sustainable meat consumption pattern: a shift in the type of meat consumption [15,21,22]. The literature shows that poultry has less of an environmental impact compared to other meat products such as beef, lamb, and pork [8]. Therefore, from the point of view of increasing sustainable food consumption, it is beneficial for consumers to purchase more poultry and less of other types of meat. To achieve this goal, we tested an in-store intervention that could make people buy more sustainable meat products, like poultry, and less of other types of meat. Since consumers purchase meat on a regular basis, this becomes a habitual purchase [23,24]. If we want to change this habitual purchase, it can be more effective to use a strategy focusing on automatic influence, rather than trying to change the rational thinking process of consumers [25,26]. Redesigning the choice environment at the point of purchase can operate as such an automatic influence [25,26].

As visual cues are omnipresent in an in-store environment, they provide an interesting opportunity to implement a redesigned environment [26]. Our vision lets us scan a wide area of our environment immediately in a very easy and fast manner, and makes us able to absorb a myriad of these environmental visual cues [27,28]. Marketers use different kinds of visual cues to increase visibility and attract attention to products. This again can increase the likelihood of purchasing a product [25,26,29,30]. Based on this, it is clear visual cues can be used to design a choice environment [26]. Still, there are several, simple cues that have been overlooked. In this paper we suggest a combined nudge to operate as a visual cue in a supermarket; an increase in the size of the display area, together with an increase in the quantity of displayed products within this display area (the same assortment of products, but offered in bigger amounts). This is a combination because the two nudges do not necessarily occur together. For example, the quantity of displayed products can be increased by stacking the products so the size of the display area remains the same. The target product in this nudging intervention is poultry, as we want to promote the consumption of meat with a lower environmental impact.

Prior research on the effectiveness of visual nudges has focused mainly on the positioning, lighting, and order of how products are displayed [31,32]. With this paper we extend the literature by using a novel nudge that combines the simple cues of display area size and quantity of displayed products. No prior research has investigated the effect on purchase behavior of this combination of two nudges in a real-life in-store environment. Furthermore, most prior research on visual nudges took place in restaurants, cafeterias in schools and hospitals or even in isolated experimental environments [31,32]. Our paper focuses on a real-life supermarket, which is a much more relevant environment to monitor the impact of a nudge on actual purchasing behavior. Another benefit of using a real-life environment is that we have collected purchasing data from actual consumers, whereas a lot of previous research relies on students [31,32].

### 1.1. Effect of Visual Cues in a Product Choice Environment

Prior research has already shown the effect of visual cues on the different aspects of decision making, like product attitude, purchase intention and even consumption [33]. For example, prior research has shown that when certain products were placed on the top of a menu, they were purchased significantly more often compared to when they were placed in the middle of the menu [34]. Visual cues can come in very different forms. Some visual cues explicitly provide some additional information and are mostly applied on the product itself, such as the color, shape, or type of lettering that is used on the package—in general, the overall look of the product [35,36]. However, visual cues are also used when designing the product choice environment. For example, the specific vertical and horizontal positioning of certain products, lighting, and table setting are visual cues that enhance product visibility by changing the environment of the product [31,32,37].

Both types of cues, which are either applied to the target product or to the surroundings of the target product, can increase the visibility of the product [31,38]. For example, a prominent color on the one hand, or the placement of a product at eye level on the other hand, increases the visibility of the product and hence attracts the customer’s attention [38]. Thus, consumers will pay more attention to products that are more visible, which again will have a positive impact on the likelihood that a product will be chosen [39]. For example, previous research showed that customers especially buy products that are located not more than 30 cm above or below eye level [38].

Besides capturing attention, increasing visibility can lead to a higher perception of the product availability [32]. If the quantity of a certain displayed product increases, it will not only become more visible, but it will also be perceived as more available [32,40]. Visual cues increasing availability can also operate as a popularity cue [41]. Consumers perceive the number of displayed products as an indication of the preference of other consumers [41]. This perceived preference can act as a social norm which can drive consumer behavior. In this manner availability also serves as a quality cue, as consumers get the impression that products with a lot of stock on the shelf must be of good quality [41,42].

### 1.2. Adapting the Choice Environment

By using visual cues, we can design a choice environment that impacts consumer behavior. In the last decade, academic research has looked thoroughly at the impact of adapting choice environment on consumer behavior [26,31,32]. Creating a certain environment that consumers are faced with is called ‘choice architecture’ [26]. Studies on choice architecture often use insights from behavioral economics to explain why choice architecture can lead consumers in a certain direction [26,31,32]. Interfering in choice architecture is often called nudging when this choice architecture “alters the behavior of consumers in a predictable way without eliminating certain options or changing the economic incentives of a certain option” [26] (p. 6). Nudging is mainly applied to promote ‘better choices’ like, for example, buying more healthy or environmentally friendly food [30,32,43]. In this paper, the ‘better choice’ is purchasing poultry.

Various nudge frameworks have been suggested to categorize different types of nudges like, for example, the MINDSPACE (Messenger, Incentives, Norms, Defaults, Salience, Priming, Affect, Commitments and Ego) framework [44] or the TIPPME (Typology of Interventions in Proximal Physical Micro-Environments) framework [45]. However, most of these frameworks are rather explorative, instrumental, and descriptive, whereas the framework of Cadario and Chandon (2018) [31] on healthy nudges is theory-based and has been empirically validated by a meta-analysis. Their classification is based on the classic tripartite distinction of mental activities: cognitively oriented (i.e., descriptive nutritional labeling, evaluative labeling, and visibility enhancements); affectively oriented (i.e., hedonic enhancements and healthy eating calls), and behaviorally oriented (i.e., convenience and size enhancements). The theoretical basis of this framework allows researchers to make predictions about the effectiveness of certain nudges and let them better understand the functionality of different nudges, especially in an in-store environment. For example, this framework takes different types of food behavior into account (food selection and food consumption) and is tailored for nudges used in an in-store environment [31].

In this paper, the nudge we propose is a combination of display area size and quantity of displayed products. Combining these two nudges provides added value compared to using stand-alone nudges because characteristics of both a cognitive-oriented nudge as well as a behavioral-oriented nudge are at play [31,46]. The implementation of behavioral nudges in a nudging intervention is important because they have the strongest impact on consumer behavior [31]. In addition to this, a combination of nudges has also been proven to be more effective than single nudges at generating effects on consumer behavior [31]. By increasing display area size and the quantity of the displayed poultry products, visual enhancements are made that could provide additional information to the consumer. As a result, visibility of poultry products is higher which can affect the perceived availability of the product, and this increased availability can send popularity cues about the product to the consumer [32,41]. This is the cognitive-oriented part of our nudge [31]. On the other hand, increasing display area size and the quantity of the displayed poultry products also enhances convenience as poultry products became the most prominent type of meat available in the display area as a result of our nudge. In respect of this, consumers can get the impression that poultry is the default option, which can have an effect on the ease of selecting this product. Because of this, it takes more effort to consider other types of meat and less effort to consider poultry for purchase, leading to an increase in convenience in favor of purchasing poultry [26]. This is the behavioral-oriented part of our nudge [31].

### 1.3. The Aim of This Paper

This paper uses a nudge based on the sensorial cue of visibility. We increased the visibility (which could increase the perceived availability and popularity) of a sustainable meat product (poultry) at the butcher counter of a supermarket by enlarging the display area size for poultry products and also the quantity of displayed poultry products. At the same time, we reduced the display area size and quantity of displayed products for meat products with a higher environmental impact such as lamb and beef. Through these methods, we want to create a shift in the type of meat consumption towards consuming more poultry and fewer other types of meat. By enlarging the display area size and increasing quantity of displayed poultry products the visibility of these products will increase; by reducing the display area size and the quantity of displayed lamb and beef products their visibility will decrease. Since visibility has been shown to affect sales of products, we hypothesize that:

**Hypothesis** **1** **(H1).**
*Increasing (decreasing) the display area size and quantity of raw meat products displayed within this display area will lead to more (fewer) sales of these meat products.*


## 2. Materials and Methods

### 2.1. Set-Up

We implemented the nudging intervention in the butcher counter of a local supermarket of a mid-sized European city. Next to the supermarket in which we implemented the nudge, we also selected a similar supermarket to serve as a control store. This design allowed us to minimize the impact of external confounding factors (for example, different store lay-outs). Both supermarkets are part of the same supermarket chain of ecological and biological products offering health benefits and reducing environmental impact, thereby targeting consumers who consider environmental concerns in their purchasing decisions. Our nudging intervention aims to increase the sales of meat products with a lower environmental impact (poultry) by changing the choice architecture of the butcher’s department of the supermarket. To do so, we enlarged the display area size and the quantity of poultry products displayed at the butcher counter.

In order to practically implement our visual cues, the display area size for poultry products was horizontally enlarged, going from 1.3 m to 1.85 m in length (an increase of 42%). The number of poultry products offered in the display area was also increased to 27 instead of 19 plates (an increase of 42%). The variety of poultry products was kept the same, but the quantity of the specific products increased. As a result of these changes, the display area size for several other meat products needed to be reduced so the total display area size could remain constant. We reduced the display area size for veal, beef, lamb, and prepared meat dishes. The display area size of veal, beef, and lamb was reduced from 80 cm to 55 cm in length (a decrease of 31%). The display area size for the prepared meat dishes was decreased from 1.3 m to 1 m (a decrease of 27%). The butcher counter also offers pork products, but the display area for these products was kept the same. An overview of the alterations in the display areas can be found in Figure 1. Figure 2 shows the display area of poultry during the intervention.

The intervention period lasted four weeks from 5 March 2018 to 31 March 2018. Right after this period, Easter took place (1 April 2018). During this period, lamb is sold more often compared to the rest of the year. Because we will always compare the results of the experimental store with the results of the control store, we are able to control the potential effect Easter had on the sales of meat. Before the start of the intervention, the store manager and the employees of the butcher counter of the supermarket received a thorough briefing about the general objectives of the intervention and the changes that would be implemented. During the intervention period, frequent visits were made to the experimental store to see if the changes in the display layout were consistently applied. A proper preparation and follow-up also contributes to the validity of this experiment [47]. During the four-week intervention period, we monitored the sales data of the different meat products that were offered by the butcher counter at both the experimental and control stores. We also included a four-week pre-measurement and a four-week post-measurement in both the experimental and control stores. The pre-measurement contained data for the four weeks from 5 February 2018 to 3 March 2018. The post-measurement also consisted of four weeks of data, from 3 to 28 April 2018. The raw data we received consisted of the total weight sold per product per day and the revenue these products generated per day.

### 2.2. Participants

The participants of this study were customers of a butchery in a supermarket located in a mid-sized European city. We cannot provide other consumer characteristics because the retailer only provided raw data on total daily sales for each meat product sold by the butcher counter.

### 2.3. Statistical Analysis

To compare the total weight sold (kg) and revenue (Euros) in the experimental store with the control store, two three-way ANOVA’s were conducted to analyze the main effects and all interactions of timing (Pre-measurement vs. Intervention vs. Post-measurement), type of store (Experimental vs. Control), and type of meat (Poultry vs. Other meat vs. Pork). Beef, veal, lamb and other meat prepared dishes were combined in a new variable; other meat. Weight and revenue were separately entered as dependent variables. The unit of analysis was the weight sold per product (e.g., chicken breast, chicken sausage, turkey tenderloin, etc.) per day and the revenue generated per product per day. Type of store, timing, meat type, and their interactions were treated as fixed factors. All data were analyzed using IBM SPSS Statistics 25 (IBM corp., Armonk, NY, USA).

## 3. Results

### 3.1. Effects on Sold Weight (Average Amount of Weight (kg) per Product per Day)

There was a significant effect of the type of store on the weight sold (*F*(1, 17,448) = 48.94, *p* < 0.001). Sales in the experimental store (mean (M) = 0.85, standard deviation (SD) = 2.17) were significantly lower than these in the control store (M = 1.1, SD = 2.56). There was also a significant effect of the type of meat on the weight sold (*F*(2, 17,448) = 641.83, *p* < 0.001). Sales of poultry (M = 1.91, SD = 3.67) were significantly higher (*p* < 0.001) than those of other meat products (M = 0.65, SD = 1.49) and significantly higher (*p* < 0.001) than the sales of pork (M = 0.41, SD = 0.84). There was no significant effect of timing on the weight sold (*F*(2, 17,448) = 1.25, *p* = 0.287). Thus, the average weight sold of all meat products together was similar before, during and after the intervention period (Before: M = 0.96, SD = 2.36; During: M = 1.02; SD = 2.54; After: M = 0.95, SD = 2.21).

We did not find a significant interaction between type of store and timing (*F*(2, 17,448) = 0.226, *p* = 0.798). There was also no difference between the type of store and the type of meat (*F*(2, 17,448) = 2.09, *p* = 0.124). We did not find a difference between timing and the type of meat (*F*(4, 17,448) = 1.79, *p* = 0.128). Also, when we looked at the interaction between the type of store, timing, and the type of meat, we did not find a significant difference (*F*(4, 17,448) = 0.85, *p* = 0.495). Although we did not find an interaction effect, when performing contrast tests we did find important significant differences, showing the effectiveness of our nudge. There was a significant increase (+13%) in the amount of poultry sold in the experimental store when we implemented the intervention (*p* < 0.05). When the nudge was removed, we saw a significant decrease (‒18%) in sales at the experimental store (*p* = 0.001). We did not find these significant differences for poultry in the control store when comparing the pre-measurement with the intervention period (*p* = 0.883) or comparing the intervention period with the post-measurement (*p* = 0.277). The significant differences found in the experimental store highlight the effectiveness of increasing the size of the display area and quantity of displayed poultry products. Despite the significant increase of sold poultry, the weight of all types of meat sold did not change significantly in the experimental store when comparing the pre-measurement with the intervention period (*p* = 0.139) and the intervention period with the post-measurement (*p* = 0.978).

We did not find a significant decrease in the amount of weight of other meat products sold in the experimental store when we implemented our nudge (*p* = 0.978). When the nudge was removed again we did not see a significant difference (*p* = 0.859). In the control store we also did not find a significant difference when comparing the pre-measurement with the intervention period (*p* = 0.258) and the intervention period with the post-measurement (*p* = 0.442). This shows that the decrease of size of the display area and quantity of displayed products did not have an effect on sales of the less sustainable meat products. Finally we did not find a significant difference in the weight of pork sold in the experimental store when we implemented the nudge (*p* = 0.841) or when the nudge was removed again (*p* = 0.738). This was expected as the size of display area and quantity of displayed pork remained constant. In the control store we also did not find a significant difference comparing the pre-measurement with the intervention period (*p* = 0.753) and the intervention period with the post-measurement (*p* = 0.599) (See Table 1 for the average weight of meat products sold).

### 3.2. Effects on Revenue (Average Revenue (Euros) per Product per Day)

We performed the same three-way ANOVA, but revenue was used in this case as a dependent variable. There was a significant effect of the type of store on the revenue (*F*(1, 17,448) = 53.96, *p* < 0.001). Revenue in the experimental store (M = 14.13, SD = 38.92) was significantly lower than revenue in the control store (M = 19.06, SD = 46.53). There was also a significant effect of the type of meat on the revenue (*F*(2, 17,448) = 507.36, *p* < 0.001). Revenue from poultry (M = 31.49, SD = 67.74) was significantly higher (*p* < 0.001) than the revenue from other meat products (M = 12.16, SD = 26.30) and also significantly higher (*p* < 0.001) than pork (M = 6.10, SD = 13.01). Once again, there was no significant effect of timing on the revenue (*F*(2, 17,448) = 1.53, *p* = 0.215). 

There was a significant interaction between the revenue by type of store and the type of meat (*F*(2, 17,448) = 4.65, *p* = 0.01). Post hoc, simple effects analysis showed that the revenue of poultry in the control store (M = 35.31, SD = 71.61) was significantly higher (*p* < 0.001) than the revenue from the experimental store (M = 27.66, SD = 63.42). The revenue from other meat products was also significantly higher (*p* < 0.001) in the control store (M = 14.35, SD = 9.99) compared to the experimental store (M = 9.99, SD = 20.38). There was no significant interaction between timing and the type of meat (*F*(4, 17,448) = 0.820, *p* = 0.512). Between timing and the type of store, there was no significant difference (*F*(2, 17,448) = 0.213, *p* = 0.808). Also when we looked at the interaction between type of store, timing, and type of meat, we did not find a significant difference (*F*(4, 17,448) = 0.92, *p* = 0.449). Nevertheless, the changes in revenues do align with our expectations. Hence, when analyzing the results via contrast tests, we again saw significant differences. 

There was a significant increase (+18%) in revenue of poultry in the experimental store when we implemented our nudge (*p* < 0.05) compared to the pre-measurement. When we removed the nudge, we saw a significant decrease in revenue in the post-measurement compared to the intervention period (*p* < 0.05). In the control store, we did not have a significant difference in revenue when the intervention period was compared with the pre-measurement (*p* = 0.614) or comparing the post-measurement with the intervention period (*p* = 0.926). The total revenue of meat sold did not significantly change in the experimental store when comparing the pre-measurement with the intervention period (*p* = 0.139) and comparing the intervention period with the post-measurement (*p* = 0.372). The revenue for other meat products did not significantly change in the experimental store when we used our nudge (*p* = 0.841) or removed it again (*p* = 0.616). In the control store there was also no significant difference between the intervention period and pre-measurement (*p* = 0.109) or between the intervention period and post-measurement, (*p* = 0.601). Finally the revenue of pork in the experimental store did not significantly change during the intervention period (*p* = 0.988) or when the nudge was removed again (*p* = 0.760). In the control store, the revenue of pork also did not change in the intervention period compared to the pre-measurement (*p* = 0.916) or during the post-measurement compared with the intervention period (*p* = 0.627) (See Table 2 for the average revenue of meat products sold). 

## 4. Discussion

The results of our field experiment showed that changing the size of the display area and the quantity of displayed products in the display area can have an effect on the sales of more sustainable meat products. When the display area size and quantity of products displayed are increased, the sales of these products increase, which gives further support of the impact that visual cues can have on consumer behavior [32,33,40]. If we take away our nudge, we notice a decrease in the sales of the more sustainable meat product. This also demonstrates that our composite nudge has a positive effect on the sales of poultry. Our nudge increased sales of the sustainable meat product when we enlarged the display size and quantity of displayed products and the sales again decreased when our nudge was removed. However we did not see an effect on sales when the display area size and quantity of products displayed of the less sustainable products were decreased. Thus, the sales of less sustainable products did not decrease. In respect of this, we can partially accept our hypothesis; when we increase the size of both display area and quantity of products displayed this has an effect on the sales of more sustainable meat, but decreasing the size had no effect on the sales of less sustainable meat products.

The lack of a significant difference in sales of the less sustainable meat products could be a result of the main walking direction of the customers towards the butchery. When following the main walking direction in the store, customers first encounter the area of the butchery where the less sustainable meat products are displayed (See Figure A1 in Appendix A). Therefore the less sustainable meat products were still visually displayed in a favorable position, which could have diminished a possible effect of the nudging intervention in the experimental store. Although there is a possible effect of the main walking direction in the experimental store, we may rule this out as a confounding factor since we compare the results of the experimental store with the results of the control store which has a very similar lay-out as the experimental store. Finally, with our nudge we also wanted to create a shift in meat consumption, rather than decreasing meat consumption in general. Based on the results, we can state that that shift has occurred as the sales of poultry increased significantly in the experimental store, but the total amount of meat sold did not change significantly.

### 4.1. Contributions

On the one hand, this paper extends the existing literature by giving additional empirical evidence that visual cues can impact consumer behavior, and more specifically, purchasing behavior. Thereby, we demonstrate that a combined nudge of size of the display area and quantity of products displayed can operate as a visual cue that can have an actual effect on purchasing behavior. On the other hand, the methodology used in this paper contributes to a large extent to the existing literature. Most prior research has used closed experimental settings, school restaurants, or hospital cafeterias as an experimental location, whereas this study took place in a real supermarket, which is much more relevant given the great external validity [31,32]. Next to a real-life and relevant experimental location, our study also examined participants who are actual customers, which again increases external validity of the reported findings over extant research. Most studies to date use student participants, which might not be representative of the general consumer.

### 4.2. Limitations and Future Research

In this experiment, we only measured purchasing behavior; we do not have an overview of the actual consumption behavior of the participants. The data were also provided as the sales of each meat product per day, which did not allow us to take a closer look at the consumer profile. We did not have data concerning price fluctuations or price changes in the stores. However by comparing our results with a control store of the same branch we can still ascribe the results to our nudge as in both the experimental and control store the same price strategy is used and thus price changes were applied equally in both stores. In our study we only used one control store, whereas multiple control stores can create a more robust control environment. The store we used to implement our nudge sells biological and ecological products and thus targets consumers who already make more sustainable decisions. We do see that even purchasing behavior of consumers who probably have a high environmental concern, can be nudged towards more sustainable food choices. As this paper used a field experiment with real sales data, we cannot determine the underlying mechanism that drives the effect. As mentioned earlier in the paper, visibility, availability, and popularity are different mechanisms that can drive this effect. Further research could tell us which underlying mechanism is driving the effect of the nudge we implemented. We used a combined nudge in this paper, so we cannot make conclusions for each nudge individually. To make conclusions for the separate nudges, several lab experiments can be set up in which the effects of each nudge can be tested.

Finally, the specific nudge we used steers consumers in a direction without deliberate thought. Consumers are more inclined to choose a product that is more visible, not realizing they are choosing the most sustainable and visible option. They probably were not aware that they chose a sustainable product but they chose that product because the display area and displayed products for poultry were more visible. To ensure that consumers make a sustainable decision in the future—even in the absence of a nudge—it could be necessary to, for example, inform or compliment them about their sustainable purchases in order to maintain performance of the more sustainable behavior. Further research can investigate the interplay between nudging and communication, where our nudge could be followed by a reward for consumers (for example, a text that says ‘Thank you for choosing the more sustainable option’).

## 5. Conclusions

The current research extends our understanding of nudging consumers towards more sustainable food choices by using a simple visual cue. Our nudge, which combined size of the display area and the quantity of products displayed, succeeded in shifting meat choices towards purchasing more sustainable meat when our nudge increased the visibility of the more sustainable meat products.

## Figures and Tables

**Figure 1 foods-08-00186-f001:**
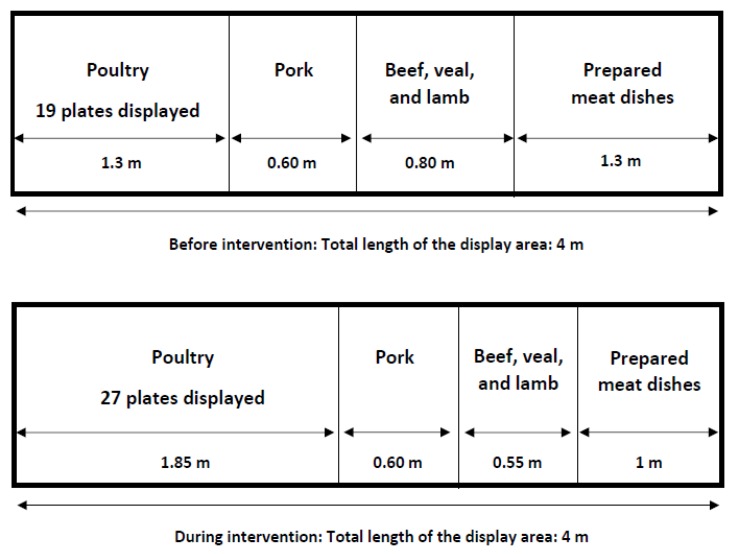
Schematic overview of the display area in the butchery before and during the intervention.

**Figure 2 foods-08-00186-f002:**
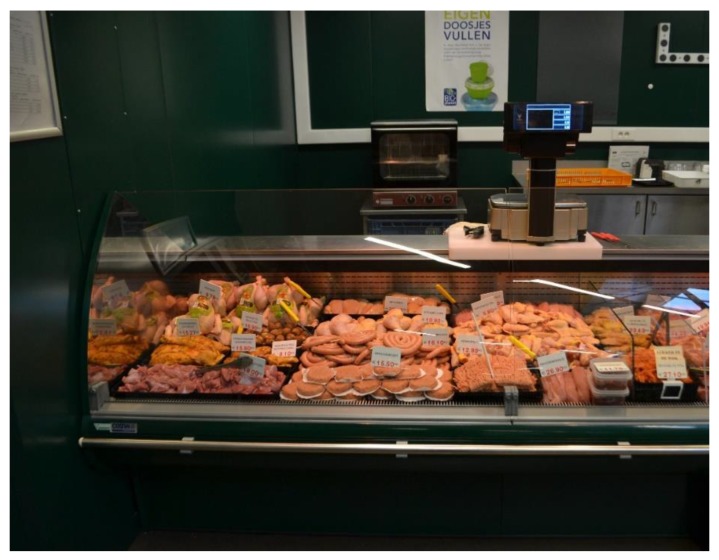
Photo of the display area of poultry in the butchery during the intervention.

**Table 1 foods-08-00186-t001:** Amount of meat (kg) sold per product per day.

Type of Store	Type of Meat	Pre-Measurement (1)	Intervention (2)	Post-Measurement (3)	Difference (1) & (2)	Difference (2) & (3)
Experimental store	Poultry	M = 1.70	M = 1.93	M = 1.58	*p* = 0.034	*p* = 0.001
		SD = 3.46	SD = 3.91	SD = 3.04		
	Other meat	M = 0.53	M = 0.53	M = 0.54	*p* = 0.978	*p* = 0.859
		SD = 1.07	SD = 1.22	SD = 0.98		
	Pork	M = 0.33	M = 0.31	M = 0.35	*p* = 0.841	*p* = 0.738
		SD = 0.70	SD = 0.70	SD = 0.74		
Control store	Poultry	M = 2.12	M = 2.14	M = 2.02	*p* = 0.883	*p* = 0.277
		SD = 3.92	SD = 3.92	SD = 3.65		
	Other meat	M = 0.72	M = 0.82	M = 0.75	*p* = 0.258	*p* = 0.442
		SD = 1.60	SD = 1.96	SD = 1.79		
	Pork	M = 0.50	M = 0.46	M = 0.53	*p* = 0.753	*p* = 0.599
		SD = 0.95	SD = 0.85	SD = 1.03		

**Table 2 foods-08-00186-t002:** Revenue (Euros) generated per product per day.

Type of Store	Type of Meat	Pre-Measurement (1)	Intervention (2)	Post-Measurement (3)	Difference (1) & (2)	Difference (2) & (3)
Experimental store	Poultry	M = 26.06	M = 30.74	M = 26.17	*p* = 0.018	*p* = 0.022
		SD = 61.18	SD = 67.65	SD = 61.06		
	Other meat	M = 9.51	M = 9.83	M = 10.63	*p* = 0.841	*p* = 0.616
		SD = 18.90	SD = 22.84	SD = 19.12		
	Pork	M = 4.58	M = 4.61	M = 5.31	*p* = 0.988	*p* = 0.760
		SD = 10.03	SD = 11.26	SD = 11.22		
Control store	Poultry	M = 34.71	M = 35.71	M = 35.52	*p* = 0.614	*p* = 0.926
		SD = 68.65	SD = 72.08	SD = 73.99		
	Other meat	M = 12.94	M = 15.47	M = 14.64	*p* = 0.109	*p* = 0.601
		SD = 26.89	SD = 34.35	SD = 31.17		
	Pork	M = 7.15	M = 6.91	M = 8.02	*p* = 0.916	*p* = 0.627
		SD = 14.63	SD = 13.94	SD = 15.70		

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
