# Peer review of "Show Me More! The Influence of Visibility on Sustainable Food Choices"

_foods, 2019, doi:10.3390/foods8060186_

Reviewer 1 Report

Overall, this is an interesting and well-written paper. I have some concerns that must be addressed, though.

Why is it interesting to look at revenue? You did not collect info on discounts, price changes etc, so fluctuating prices make it difficult to make any conclusions based on these data.

Mixing two nudges is interesting, but now you cannot say anything about the effect of each nudge, right? Also, how certain are you that shops complied with the number of products displayed and refilled sold products?

l. 123: Be more specific.

l. 132-133: Default does not necessarily equals convenience in this context. Also, you have not specified types of products, they may differ in difficulties of preparation.

l. 148: The hypothesis does not cover all aspects of both nudges.

Figure 1. Please also provide the number of products in the figure.

Please indicate whether the study period covered religious days (e.g. Easter) or other special days where certain foods are traditionally eaten.

Did you obtain ethical approval to conduct the study and GDPR approval to store the data?

l. 209-210: Why not? How can it then be significant according to the above?

Merge table 1 and 2 and add p values. The same for table 3 and 4.

l. 288. Insert “composite” or similar in front of nudge.

l. 314-315. I do not think you can say anything about sustainability based on this study. You looked sized of display area and quantity of displayed products. Nothing in the study lead the consumer to think about sustainability. The fact that poultry products are more sustainable is not at argument in it self. Such products may also have lower fat content, but this is not mentioned either.

l. 321. Earlier you called it organic. Be consistent.

l. 334. Are poultry products really sustainable?? Or just more sustainable than red meat?

Author Response

Dear Editor,

You can find the point-by-point responses in the attachment.

Kind regards

Reviewer 2 Report

The paper deals with a very interesting topic, it was pleasant to read and well written. Overall I find no major problems related with the paper, although some minor checks (mostly typos) are required:

- line 115. Please amend MINDSPACE instead of MINDSCPACE

- line 116-117. According to references, the paper from Cadario and Chandon is from 2017. Please revise.

- line 178. This figure should be improved. Could you provide a real picture of the counters? It might be of interest for readers. If not, the quality should be improved ("red lines" mostly and the sharpness of the writings).

- line 201. This is just a curiosity. Did you try also to analyze the "other meat" separately?

- line 309. Since you don't have any information regarding socio demographics characteristics of your sample, how can you state that you examined "adult participants"? I might add some conditional verb. 

- line 310. This line regarding students is very strong. Also here, it might be an option to use a conditional verb.

- line 324. "towards more sustainable food choices." After this dot, I think there is a double-space. 

Author Response

(The authors gave the same response as above.)
